# Molecular Insights into Pleural Mesothelioma: Unveiling Pathogenic Mechanisms and Therapeutic Opportunities

**DOI:** 10.3390/diagnostics15111323

**Published:** 2025-05-24

**Authors:** Teodora Zahiu, Carmen Mihaela Mihu, Bianca A. Bosca, Mariana Mărginean, Lavinia Patricia Mocan, Roxana-Adelina Ștefan, Rada Teodora Suflețel, Carina Mihu, Carmen Stanca Melincovici

**Affiliations:** 1Radiology and Imaging Department, County Emergency Hospital, 400006 Cluj-Napoca, Romania; teodorazahiu@gmail.com; 2Department of Morpho-Functional Sciences, Discipline of Histology, “Iuliu Hațieganu” University of Medicine and Pharmacy, 400012 Cluj-Napoca, Romania; bianca.bosca@umfcluj.ro (B.A.B.); mariana.marginean@umfcluj.ro (M.M.); trica.lavinia@umfcluj.ro (L.P.M.); roxanalupean92@gmail.com (R.-A.Ș.); sufletel_rada@yahoo.com (R.T.S.); carmen.melincovici@umfcluj.ro (C.S.M.); 3NovoGyn, Dostoievski 16, 400000 Cluj-Napoca, Romania; carina.mihu@umfcluj.ro

**Keywords:** BAP1, MTAP, asbestos, pleural mesothelioma, immunohistochemical markers, immunohistochemistry, inflammatory microenvironment, diagnosis, prognosis

## Abstract

Pleural mesothelioma (PM) is a rare disease, which is going to be a global medical concern in the 21st century, because of its aggressiveness, late diagnosis, and insufficient therapies. This review seeks to enhance the comprehension of medical professionals regarding the risk factors and environmental influences that contribute to the development of the disease, as well as its underlying mechanisms. In addition, we aim to provide a schematic yet thorough overview of diagnostic techniques in PM, emphasizing the significance of the immunohistochemical markers BAP1 and MTAP, with the latter serving as an almost ideal surrogate for the gold-standard diagnostic approach, FISH p16/CDKN2A deletion. The scientific world is grappling with BAP1, MTAP, and the tumour inflammatory microenvironment, because they are the key for personalized treatments and palliative care in this disease. Considering that the survival rate for patients with PM seldom surpasses five years, every moment is significant. Therefore, our article also highlights recent advancements in clinical assessments related to prognostic scoring and treatment options. PM is a complex disease, with gradual progression over decades, which requires further investigation covering the prevention, mutations, diagnosis and treatment.

## 1. Introduction

Malignant mesothelioma (MM) is a rare but aggressive tumor which arises from mesothelial lining cells, mainly caused by exposure to asbestos. In the majority of cases (80–85%), MM involves the pleura, a structure that covers the lungs and lines the interior walls of the thorax [1,2]. The peritoneum is the second point of origin, but other membranes like the pericardium and, rarely, the testicular tunica vaginalis, may also be involved [3].

Malignant pleural mesothelioma rates and deaths have increased in recent decades in developed nations, especially among those exposed to asbestos at work [4]. The pathology is associated with poor prognosis, with an average survival rate in untreated patients between 7–10 months [2,5]. Recent studies [3,6] reported that only 12% of patients reach 3 years after diagnosis. Palliative care has improved patients’ quality of life and extended average survival post diagnosis to 9 to 12 months [7].

The high mortality rate of pleural mesothelioma (PM) and the ineffectiveness of palliative treatments highlight the urgent need for innovative diagnostic, prognostic, and therapeutic approaches [6,7,8,9].

This paper will focus on the epidemiology, risk factors, pathogenesis, diagnostics, molecular features, and potential therapies for PM.

## 2. Epidemiology

PM primarily affects individuals aged 40 to 60 but can also occur in younger populations depending on exposure. Diagnosis usually presents unilaterally [10,11]. Before 1960, PM was considered rare, but that year saw 33 cases documented among South African workers exposed to blue asbestos (crocidolite), marking a turning point. Today, PM is recognized as an occupational disease [12].

Due to professional exposure, the majority of patients (70%) are men [9,13], but paraprofessional exposure can also affect women and children [8,9], considering the possibility of indirect impacts resulting from the inhalation of particles present on the worker’s clothing or personal items. The risk of PM is directly proportional to cumulative asbestos exposure, duration, and latency period [13].

Most countries have laws and regulations regarding the use of asbestos, but developing countries are still at risk in terms of public health, with the incidence and mortality caused by mesothelioma being on the rise [7,14]. Between 1994 and 2016, PM caused 50,000 deaths in Europe, which represents 56% of the PM related-deaths worldwide [7]. In recent years, approximately 2500–3200 new cases per year [15,16] have been diagnosed in the United States, and approximately 38,400 new cases per year [3] have been diagnosed worldwide.

Developed countries have restricted asbestos use and promoted alternatives, while developing nations, particularly China and India, account for about 50% of global consumption. Russia, Kazakhstan, China, Brazil, and Zimbabwe still produce and export asbestos [7]. As a result, WHO estimates that approximately 125 million people are currently being exposed to asbestos in their homes or workplaces [10].

## 3. Risk Factors and Pathogenesis

Asbestos has been used for over 5000 years in various industries due to its unique physical and electrochemical properties, including electrical insulation, thermal insulation, and durability [7]. This material was widely used in the industry in the last century and is still found in the structure of many buildings today [7]. Asbestos-related diseases, especially lung cancers, present significant public health challenges due to their long latency. Research indicates no safe exposure threshold, with a clear dose–response relationship between exposure levels and lung cancer risk, which increases over time with greater exposure [7]. In some areas, PM can be caused by exposure to other bio fibres such as erionite, zeolite, ophiolite, and fluoro-edenite [10].

The carcinogenic potential of a mineral is influenced by its physical characteristics, including diameter, length, and biodegradability. The WHO indicates that fibres under 3 μm in diameter and over 5 μm in length, with a length-to-diameter ratio greater than 3, pose the highest mesothelioma risk due to their ability to penetrate the lung and pleura, evading removal [17,18,19].

Macrophages’ inability to fully engulf elongated asbestos fibres leads to frustrated phagocytosis, triggering chronic inflammation that creates a “mutagenic microenvironment” and causes DNA damage in pleural mesothelial cells [20].

The inadequate internalization of asbestos fibres and failed phagocytosis lead to inflammatory cell activation and the production of cytokines, such as tumor necrosis factor alpha (TNF*α*) and fibroblast growth factor (FGF). These cells also generate free radicals through macrophage-mediated reactive oxygen and nitrogen species, causing DNA damage and genomic instability, which may result in uncontrolled cell proliferation and the evasion of apoptosis [18,19,21,22].

Iron plays a key role in the toxicity of asbestos, especially in the formation of asbestos bodies (ABs) in the lungs, due to asbestos fibres’ ability to adsorb iron from their surroundings. In healthy individuals, maintaining cellular iron balance is crucial. Asbestos exposure lowers intracellular iron levels by binding the metal with fiber surfaces, disrupting essential cellular functions and causing cell death or apoptosis [23,24]. The mechanism of asbestos body formation and the accumulation of endogenous iron, ferritin, and mucopolysaccharides in lung tissue remain unclear [23,25].

Typically, after an injury, cells can repair their DNA. However, if the damage is too severe, they will undergo apoptosis. However, the persistence of the asbestos fibres and the formation of ABs maintain chronic inflammation and DNA-damaging oxidative stress [26,27]. Asbestos fibres increase ferritin heavy chain expression, causing iron accumulation and localized overload, which raises hydroxyl radical production. Thereby, asbestos’s role in carcinogenesis is affected by the amount and oxidation state of iron. Some authors suggest that surface Fe^3+^ ions are responsible for the carcinogenicity of asbestos [24,28]. However, the amount of iron in different types of asbestos is not enough to predict the degree of carcinogenesis [29].

Genetic predispositions affect mesothelioma risk [30], but exposure also impacts entire families when one member works in the asbestos industry.

Many other fibres that resemble asbestos fibres (e.g., non-asbestos crystalline fibres) are considered risk factors in PM development. Experimental studies [31,32,33] have confirmed that erionite is carcinogenic, causing lung issues like pleural fibrosis and mesothelioma [34]. Similarly, carbon nanotubes, used in medical and industrial applications due to their asbestos-like properties, may also pose a future risk for pleural mesothelioma [34]. In vitro and in vivo studies on rats [35,36] have shown the cytotoxic effects of these particles, but determining their pathogenesis in the general population is challenging.

PM begins in the parietal pleura as nodular formations that eventually merge into a continuous mass affecting both pleural layers, containing fibrous plaques [7]. Asbestos fibres can adsorb iron, leading to the incorporation of mineral fibres with ferritin and hemosiderin, resulting in ferruginous-coated fibres known as ABs. These brown nodules, rich in iron-containing proteins and acid mucopolysaccharides, are key indicators of asbestos exposure and may explain the toxic and carcinogenic effects of asbestosis [12,25].

Distant metastasis occurs late and is most commonly found in the lung, liver, kidney, adrenal gland, and, in rare cases, bone and spleen [12].

## 4. Diagnostic Methods

Exposure to asbestos can result in various clinical symptoms, including unilateral pleurisy, dry cough, hemoptysis, dysphagia, night sweats, clubbing of fingers, ascites, superior vena cava syndrome, Horner’s syndrome, laryngeal nerve paralysis, and paraneoplastic syndrome, among others. Many of these symptoms develop progressively and may mimic malignant infiltration. Given their non-specific characteristics, these symptoms do not serve as diagnostic criteria for pleural mesothelioma, even in cases of confirmed asbestos exposure [7,11,12,37,38,39].

The primary diagnostic tool is thoracic computer tomography (CT) with contrast, followed by CT-guided core biopsy or video-assisted thoracoscopic (VAT) pleural biopsy [5,40]. Magnetic resonance imaging (MRI) and 18F-fluorodeoxyglucose-positron emission tomography (FDG-PET) are also used to evaluate tumor resectability, invasion extent and treatment response [3,5]. When cytological analysis is inconclusive or ultrasound-guided pleural fluid aspiration is unfeasible, FDG-PET/CT imaging can effectively distinguish between benign and malignant conditions and locate optimal biopsy sites [3,7].

A novel imaging technique using radiomics signatures may differentiate malignant from benign lesions [41]. The study found that CT-identified malignant lesions showed increased thickness, while IRM-detected lesions had nodular contours. Ultrasound (US) is also useful for managing pleural masses, aiding in evacuation punctures for pleurisy and allowing for percutaneous pleural biopsies with US or CT guidance [40].

Imaging studies are essential for diagnosing PM, especially given that 99% of mesotheliomas present diffusely rather than locally. A thorough clinical evaluation combined with imaging techniques is crucial for accurate diagnosis and risk stratification [42,43,44]. Furthermore, in the few cases when the location is solitary, PM has a different pathogenesis and presents as less aggressive in terms of clinical manifestations [45,46], or it can be seen as an early stage of diffuse pleural PM [47].

Histopathological analysis identifies three types of PM with prognostic significance: epithelioid (60%), sarcomatoid (20%), and biphasic (20%). Each type has subtypes based on architectural patterns (tubulopapillary, trabecular, adenomatoid, solid, micropapillary) and cytological features (rhabdoid, deciduoid, small cell, clear cell, signet ring, lymphohistiocytoid, transitional, pleomorphic), with stromal characteristics including myxoid, desmoplastic, and heterologous differentiation. The latest WHO Classification of Pleural Tumors [48] defines biphasic mesothelioma as having both epithelioid and sarcomatoid patterns, with each component needing to be at least 10% in resection specimens, regardless of their proportions in small samples. Furthermore, the line between diffuse and localized mesotheliomas is well established, because the latter has a better prognosis. Alongside the latest classification, a new entity, mesothelioma in situ, has been added in the benign and preinvasive mesothelial tumors subsection, identified by the loss of BAP1 (BRCA-associated protein 1) and/or MTAP (methylthioadenosine phosphorylase) expression by immunohistochemistry (IHC) and/or p16/CDKN2A homozygous deletion detected by FISH (fluorescence in situ hybridization) [48,49].

Patients with an epithelioid pattern of PM have a much better prognosis than those with sarcomatoid or biphasic patterns [5,7,41].

In the early stages of the disease, localized peribronchiolar fibrosis occurs, progressing to involve the interalveolar septa and adjacent bronchioles, ultimately leading to diffuse interstitial lung fibrosis [34].

Histopathological findings in patients undergoing extended pleurectomy/decortication or extrapleural pneumonectomy correlate with preoperative pleural specimens [5]. However, there is no consensus on the optimal number of biopsies needed to characterize all tumor types in a patient. Despite imaging advancements, thoracoscopic biopsies from various pleural cavity regions are still recommended for diagnosis [5].

The 2015 WHO Histological Classification of Mesotheliomas [50] cautions against using only a cytological analysis of pleural fluid for diagnosis due to a high risk of false negatives. Epithelial cells are easily desquamated into pleural fluid, while sarcomatoid cells are not, complicating the identification of epithelioid or biphasic subtypes [3,7]. When biopsy samples are unavailable due to non-compliance or technical issues, clinicians should guide the evaluation of pleural cytology, biochemical markers, clinical symptoms, and imaging [3].

Respiratory sample changes occur late in disease progression, making functional pulmonary tests ineffective for screening. In contrast, MRI is more sensitive than spirometry in tracking progression in post-surgical patients [51].

FISH testing identifies diagnostic and prognostic biomarkers by detecting the loss of p16/CDKN2A from 9p21 deletion, showing 100% specificity for malignant mesothelial cell proliferation. However, its sensitivity for pleural mesothelioma ranges from 48% to 88% [1]. Despite its high specificity for mesothelial cancers [1,52], FISH’s cost and limited availability make IHC the preferred diagnostic method. However, there are no definitive markers for PM, so a panel approach with multiple antibodies is recommended. IHC typically starts with a targeted panel of mesothelial markers to confirm the lesion’s mesothelial origin.

In the epithelioid variant of PM, the following markers should be positive, indicating the mesothelial cell proliferation: calretinin, HBME1, CK5/6, WT-1 (Wilms tumor antigen-1), EMA (epithelial membrane antigen), podoplanin (D2-40), thrombomodulin, and mesothelin [1,52,53]. Aiming to exclude epithelial cell proliferation, the most commonly negative markers should be pCEA (polyclonal carcinoembryonic antigen), MOC31 and claudin-4 [1,41,52]. These markers help differentiate neoplasms of epithelial origin (e.g., adenocarcinoma or lung squamous carcinoma) from those of mesothelial origin [1,41,52]. Biological markers such as absent BAP1 immunoexpression, loss of MTAP, and the homozygous deletion of CDKN2A via FISH indicate lesion malignancy. While markers like 5-hmC and elevated enhancer of zeste 2 polycomb repressive complex 2 (EZH2) expression show promise, they are not standard due to insufficient sensitivity and specificity [1].

Sarcomatoid PM typically tests negative for markers like the loss of CDKN2A, calretinin, WT-1, mesothelin, and Claudin-4, but positive for pancytokeratin, cytokeratin 7, GATA3, AE1/AE3, CAM5.2, and EMA [54,55,56]. A negative result using an immunohistochemical panel with BAP1 and Claudin-4 can distinguish sarcomatoid PM from sarcomatoid carcinoma with 100% specificity [55]. Additionally, GATA3 presence helps differentiate sarcomatoid mesothelioma from lung sarcomatoid carcinoma, with over 70% sensitivity and about 80% specificity [57].

In tumors with a predominantly malignant morphological appearance, the confirmation of mesothelial origin is sufficient [1]. In the rest of the cases, IHC is used to distinguish between benign and malignant pathologies [1,5,6,11]. Reactive mesothelial proliferation can resemble PM and even peritoneal carcinomatosis, influenced by factors like infections, pulmonary infarction, trauma, autoimmune diseases and adverse drug reactions [3]. PM in situ can be a precursor for diffuse disease, which can be difficult to distinguish from reactive/atypical mesothelial proliferations [5].

In the absence of invasion, the loss of BAP1 immunoexpression, as well as the presence of the homozygous deletion of p16/CDKN2A by FISH, or the loss of MTAP expression by IHC, allows for the differentiation of malignant vs. benign lesions [7].

## 5. MTAP and BAP1—Molecular Biomarkers for Diagnosting and Prognosting PM

MTAP (methylthioadenosine phosphorylase) and BAP1 (BRCA-associated protein 1) are tumor suppressor genes studied for their diagnostic and prognostic value in PM. Suppressor gene inactivation in malignant mesothelial cell proliferation can be detected using FISH and IHC techniques.

The IHC method detects BAP1 loss with 100% specificity. Its sensitivity varies by mesothelioma subtype and probe used, with histological samples showing 66% to 76% sensitivity, while cytological samples range from 60% to 68% [58,59]. A contrasting study in Denmark [60] found sensitivity values of 71.43% for cytological samples and 55% for histological samples. However, it had limitations, including a small patient cohort and the lack of the gold-standard FISH technique.

Meanwhile, negative MTAP staining in PM achieves a specificity of 100% and a sensitivity of 38–44% for cytological samples and 42–55% for histological ones [58,59,60]. It remains to be seen whether cytology can be used with satisfactory diagnostic accuracy.

The goal in diagnosing PM is to achieve nearly 100% sensitivity using the immunohistochemical markers MTAP and BAP1. This raises the question: Can we obtain statistically significant values from cytological samples as effectively as from histological samples?

### 5.1. MTAP’s Function as a Tumor Suppressor Gene

MTAP, situated close to the CDKN2A tumor suppressor gene on chromosome 9p21, might also act as a tumor suppressor and could be significant in the diagnosis of PM [61]; see Figure 1. The homozygous deletion of P16/CDKN2A is present in 75% of PM and is associated with a more aggressive disease [62]. It is frequently associated with the homozygous co-deletion of MTAP, which can be identified in both histological and cytological pleural samples, with the former having a higher sensitivity (45–85% vs. 56–79%) [63].

Hamasaki et al. revealed an excellent kappa coefficient of 0.8 by correlating the homozygous deletion of the 9p21 locus (P16/CDKN2A) detected by the FISH technique in pleural cytological samples of mesothelioma, with the immunohistochemical cytoplasmic loss of MTAP. It is crucial to recognize that the primary cause of these discrepancies is not due to technical deficiencies, but rather biological variations. The study effectively concluded that the loss of MTAP serves as a more appropriate surrogate than a direct substitute [64]. Consequently, the study of the MTAP protein by IHC methods could be a surrogate for the FISH technique for determining p16/CDKN2A deletion, with the lack of MTAP immunoexpression suggesting p16/CDKN2A deletion [7,63]. The sensitivity of the diagnosis can decrease considerably to 43–65% when the sarcomatoid subtype is taken into consideration, but the specificity remains 100% [59,64,65]. The loss of MTAP expression identified by IHC is not necessarily a superior marker compared to CDKN2A deletion identified by FISH. However, when combined with another IHC marker like BAP1, these indicators improve the diagnosis and assessment of PM. The IHC technique, which detects the absence of MTAP and BAP1 proteins in biopsy samples, is preferred over the more expensive and complex FISH, recommended only for complex cases needing further validation [1,66].

The MTAP gene, which is expressed in all cells in the body, is critical in preserving the AMP synthesis pathway [67] (Figure 1). It encodes MTAP, an essential enzyme in the cellular rescue of adenosine and methionine that cleaves MTA metabolite (5′-dideoxy-5′-methylthioadenosine), a product of polyamine biosynthesis, into adenine and MTR-1-P (methylthioribose-1-phosphate). Adenine is then converted into AMP with the help of PRPP enzyme, thus blocking the conversion into toxic nucleotides. MTR-1-P is then transformed into methionine [67]. MTAP expression is reduced in several cancers, including leukemia, lymphomas, mesotheliomas, lung, and pancreatic cancers [68]. MTAP negative tumors are sensitive to de novo AMP synthesis inhibitors like L-alanosine [69], but the benefits of these therapies in PM contexts remain unclear [62,69] (Figure 1).

The absence of MTAP expression may contribute to carcinogenesis through altered polyamine metabolism [70]. Polyamines, essential for cellular growth and proliferation, are often elevated in tumors. The enzyme Ornithine Decarboxylase (ODC) regulates polyamine synthesis by converting ornithine to putrescine. Loss of the MTAP gene up-regulates ODC, increasing both ODC and polyamine production, which promotes cellular proliferation [68,71] (Figure 1). Busacca and colleagues found that the lack of MTAP expression is a negative prognostic indicator for PM progression, with MTAP-deficient patients having significantly lower overall survival than those with wild-type MTAP [72].

MTAP deficiency causes MTA to accumulate in cells, which is excreted when excessive. MTA is a potent inhibitor of the PRMT5 (protein arginine methyltransferase 5) and MAT2A (metabolic enzyme methionine adenosyltransferase II alpha) pathways, crucial for producing SAM (S-adenosylmethionine), a key substrate for PRMT5. Consequently, in cancers with MTAP loss, increased intracellular MTA leads to reduced MAT2A levels and decreased PRMT5 activity [73,74].

Furthermore, these deficiencies extend to PRMT5 protein co-complexes [74,75]. PRMT5 regulates gene expression, ribosomal biogenesis, protein translation, and mRNA splicing; its complete inactivation is not compatible with cell survival in most cell lines [76,77] (Figure 1).

Consequently, the primary area of scientific inquiry at present is the PRMT5-MTA complex and the potential methods for its inhibition [78]. Given its involvement in metabolic processes, there are numerous potential targets for pharmacological intervention, prompting the investigation of pharmaceutical combinations that incorporate both MTA inhibitors and purine analogs [79]. Further elaboration will be provided in the section dedicated to targeted therapies.

### 5.2. BAP1 Roles in Pathogenesis

#### 5.2.1. BAP1’s Cellular Roles in Cancer

BAP1 is a tumor suppressor gene located on chromosome 3 at 3p21.1 [80]. Identified by Jensen et al. in 1998 [81], it consists of 17 exons and encodes a 729-amino acid protein [82] (Figure 2). BAP1 functions as a carboxy-terminal hydrolase in the nucleus, cleaving ubiquitin–substrate bonds, particularly targeting UCH. It is part of the deubiquitinating enzyme (DUB) subfamily and features two main functional domains and potential binding partners [83,84].

Originally identified as a protein linked to BRCA1 (breast cancer gene 1), this entity is now recognized as a DUB. It features a UCH domain (1–240 aa) and a C-terminal UCHL5/UCH37-like domain (ULD, 640–710 aa), which includes a ubiquitin-like domain and two nuclear localization signals, separated by an intermediate segment of about 395 amino acids [85]. These domains facilitate interactions with various proteins, including HCF1 (host cell factor 1), FOXK1/2 (forkhead transcription factors), the BRCA1/BARD1 complex, ASXL1/2 (additional sex comb like 1 or 2), KDM1B, and YY1—Ying Yang 1 transcriptional repressor [84,85,86,87,88].

BAP1 is a versatile protein essential for regulating the cell cycle, cell differentiation and proliferation, DNA repair, gene expression, chromatin remodeling, metabolism, programmed cell death, and immune system functions [63,83,86,87] (Figure 3).

The loss of BAP1 expression can occur due to genetic anomalies like chromosomal deletions and mutations, mainly heterozygous or point mutations [89]. BAP1 mutations are present in 50–60% of PM cases, ranging from a single nucleotide to large deletions [84,86]. The ’BAP1 Malignant Syndrome’ was recently defined, and it demonstrates a significant correlation between germline mutations and cancer susceptibility, particularly melanoma, uveal cancer, PM, and renal carcinoma [90,91].

Considering that BAP1 mutations are highly tumorigenic, and that PM is an environment-dependent pathology induced by asbestosis, the importance of the association between environmental factors and genetic material cannot be overstated [86]. Research studies conducted by Napolitano [92] and Xu [93] found that BAP1 heterozygous rats had double the incidence and a faster progression of mesothelioma compared to wild-type BAP1 homozygous rats, considering environmental influences.

The BAP1 mutation exhibits autosomal dominant inheritance with incomplete penetrance and is often sporadic [83]. BAP1 germline variants have a prevalence of about 7.7% [91,94], indicating a genetic link to PM development [94]. According to Knudson’s “two-hits” hypothesis [95], affected individuals have one non-functional gene copy, while the second is inactivated over time by environmental factors like asbestos exposure, increasing their risk for PM [83].

The diagnostic sensitivity of BAP1 for pleural mesothelioma exhibits significant variation across histological subtypes, ranging from 56% to 81% for epithelioid mesothelioma and from 0% to 63% for sarcomatoid mesothelioma [96]. Furthermore, the specificity of BAP1 approaches 100% in both cytological and biopsy specimens across all three histological subtypes. It is important to note that acquiring sarcomatoid-type cells from pleural fluid is particularly challenging, and studies typically involve cohorts with a limited number of sarcomatoid subtype cases [58,59,60,65,97,98,99]. Immunohistochemical analysis indicates a loss of nuclear BAP1 expression in tumor cells, with positive staining in internal control cells like inflammatory and stromal cells [63]. This loss of the BAP1 expression occurs in both sporadic and familial PM, with the latter being linked to germline BAP1 mutations [63]. Truncating mutations and the secretion of a mutant BAP1 protein can occur in sporadic PM [91].

The absence of BAP1 staining generally indicates PM, but its presence does not rule out this diagnosis [84]. Individuals with a BAP1 mutation may still have a functional allele, leading to positive nuclear immunostaining [91]. Loss of BAP1 protein synthesis is shown by the loss of nuclear and cytoplasmic signals in IHC. Truncating BAP1 mutations can cause dysfunctional protein accumulation in the cytoplasm, forming amyloid, and resulting in cytoplasmic staining and negative nuclear staining in IHC [100,101,102].

Nevertheless, genetic modifications of the BAP1 gene in PM lead to a lack of nuclear staining, which distinguishes malignant from benign proliferations. However, IHC cannot identify genetic carriers, necessitating genotyping for further analysis.

The outcome studies attempting to highlight the prognostic value of BAP1 in PM are controversial [63,83,86].

When compared to other PM biomarkers, some studies demonstrated that the loss of BAP1 expression was associated with a young age of onset and a better average survival [26]. Furthermore, patients with BAP1 germline mutations are more likely to be female, having a younger age of onset, an epithelioid histological type, better survival, and a higher response rate to chemotherapy [86,103]. Several studies were able to demonstrate even a positive prognosis of the loss of the BAP1 expression in PM, with an increase in overall survival (OS) [104,105,106,107].

On the other hand, El-Din et al. [103] found a significant link between the BAP1 mutation and poor outcomes in PM, with affected patients experiencing faster disease progression and more frequent distant metastases to the brain, bone, and liver. Pulford et al. [108] obtained similar results regarding the BAP1 loss of immunoexpression negative prognosis.

Besides, some studies found no statistically significant link between BAP1 expression loss and survival rate [109,110].

In addition, a study by Carbone et al. [111] presents a nine-generation genealogical tree related to BAP1 cancer syndrome, highlighting the connection between genetic factors and environmental influences in cancer development [112].

It is considered that the BAP1 Malignant Syndrome is insufficiently investigated, recognized, and reported, being much more frequent than originally thought [83]. The novel approach focuses on identifying individuals and their family members with germline BAP1 mutations to assess the risk of PM and BAP1 Malignant Syndrome, which is essential for developing effective monitoring strategies and optimizing medical care [83,86]. It is desired that IHC will become a widely used screening and diagnostic method, and patients presenting a complete loss of the BAP1 nuclear expression will benefit from genetic sequencing [80,83,101].

BAP1’s increased sensitivity to the epithelioid subtype, compared to the sarcomatoid variant, is particularly useful for estimating the ratio of components in the biphasic subtype [16]. DeRienzo et al. analyzed BAP1 staining patterns, categorizing them as nuclear, cytoplasmic, absent nuclear–cytoplasmic, and combinations. They found significant correlations between these patterns and patients’ phenotypes, prognoses, and ages. Notably, a combination of positive nuclear staining, male gender, and sarcomatoid tumors was present in 36% of the samples [16].

#### 5.2.2. BAP1 as a Component of Multiprotein Complexes Involved in Cell-Cycle Control

BAP1 is present within multiprotein complexes that associate it with chromatin-related proteins, including transcription factors and regulators of the cell cycle. This association is enhanced by its localization in the nucleus, enabling BAP1 to participate in various cellular processes, such as cell proliferation, differentiation, apoptosis and so on [113].

The interaction between BAP1 and HCF-1, a chromatin-associated transcriptional cofactor, plays a role in cell-cycle control, determining cell progression from the G1 to the S-phase. BAP1 binds to HCF-1 and deubiquitinates it, enhancing its stability and preventing degradation. This interaction promotes E2F gene expression, essential for the G1/S transition. Without BAP1, cells remain arrested in the G1 phase [87].

Furthermore, the BAP1/HCF-1 complex influences cell-cycle regulation and metabolism by interacting with enzymes like OGT (O-linked N-acetylglucosamine transferase) and transcription factors such as YY1 and FOXK1/2, resulting in diverse multiprotein complexes [86] (Figure 3).

BAP1 plays a key role in gluconeogenesis and cell proliferation via the BAP1/HCF-1/OGT complex, which interacts with the COX7C promoter that encodes mitochondrial respiratory chain components [86,87,114].

BAP1 is essential for DNA repair, functioning in both BRCA1-dependent and independent mechanisms. It interacts with BRCA1-associated RING domain protein 1 (BARD1) to form the BRCA1/BARD1 complex, inhibiting a widely expressed E3 ligase and affecting the cellular response to DNA damage. Research is ongoing into BAP1’s BRCA1-independent mechanisms [86,115].

#### 5.2.3. BAP1’s Role in Deubiquitination

Ubiquitin is a 76-aa polypeptide that interacts with protein substrates via ubiquitylation, primarily facilitated by the ubiquitin-activating enzyme (E1), resulting in effects like targeted degradation, altered localization, and changes in protein–protein interactions.

DUBs enhance protein stability, functionality, and distribution by removing ubiquitin from substrates. Since protein ubiquitylation is reversible, DUBs play a crucial role in maintaining protein homeostasis [116]. Through its catalytic activity as a DUB, nuclear BAP1 influences various cellular processes, including nuclear chromatin modifications [84,86]. BAP1 forms the polycomb group repressive deubiquitinase complex (PR-DUB) with ASXL1/2 in the nucleus. Its catalytic activity deubiquitinates histones H2Aub (ubiquitinated H2A) and HCF-1, altering chromatin structure and stabilizing protein complexes that regulate the cell cycle, DNA repair, and apoptosis [84,86,87,117].

Gene-sequence alterations that reduce BAP1 expression modify the BAP1-ASXL1/2 complex, increasing H2Aub and disrupting cell-cycle progression [86,87]. Histone deacetylase inhibitors (HDAC) may provide therapeutic benefits [80].

#### 5.2.4. The Role of BAP1 in Malignant Cell Metabolism

Understanding BAP1’s role in the metabolic processes of cancer cells is crucial for predicting treatment efficacy. The following sections will explore key sub-themes, recognizing that many aspects of this topic are still under-researched.

BAP1 interferes with glucose and cholesterol metabolism by acting on both exocrine pancreas cells and hepatocytes. In the pancreas, the loss of BAP1 expression reduces mitochondrial proteins like COX6c, Trap1, and Suox, while increasing pancreatitis biomarkers. This disrupts pancreatic acinar cell homeostasis, resulting in acinar cell atrophy and loss [86,114]. On the other hand, at the hepatic level, BAP1 loss leads to increased cholesterol biosynthesis, reduced gluconeogenesis, and disrupted lipid homeostasis, resulting in decreased intra-hepatic lipid deposits, hypercholesterolemia, and hypoglycemia [86,101] (Figure 4).

In malignant cells, the Warburg effect is characterized by an increase in aerobic glycolysis and lactate production [84,86]. Bononi and collaborators [118] demonstrated that human fibroblasts of the carriers of heterozygous germline BAP1 mutations (BAP1+/−) exhibit a Warburg effect-like metabolic alteration, with mitochondrial respiration suppressed [118].

#### 5.2.5. The Role of BAP1 in Programmed Cell Death (Regulation of Cell Death)

Malignant cells adapt their metabolism and cell-death mechanisms to survive [86]. Managing metabolic stress is vital, as it can cause cell death, which may also be triggered by environmental factors like nutrient scarcity and oxygen deprivation. Initially, there was a correlation between the BAP1 nuclear localization and the anticancer properties, suggesting that DUB activity controls nuclear targets involved in gene transcription. Carbone and colleagues found that cytoplasmic BAP1 in fibroblasts is localized to the endoplasmic reticulum (ER), where it regulates intracellular calcium release and initiates apoptosis [86,101,113,114]. BAP1 facilitates the deubiquitination of the type 3 inositol-1, 4, 5-triphosphate receptor (IP3R3), stabilizing the IP3R3-ER channel essential for Ca^2+^ release into the cytosol. This calcium is then taken up by mitochondria, promoting apoptosis [119].

The loss of BAP1 expression leads to decreased stability of IP3R3 and alters the apoptosis pathway, refs. [84,86] primarily impacting fibroblasts and mesothelial cells [101]. This mechanism plays a crucial role in the carcinogenesis of uveal and cutaneous melanoma, both part of the BAP1 Malignant Syndrome [87].

Zhang et al. [120] highlighted BAP1’s role in cell-cycle regulation through ferroptosis, an iron-dependent form of programmed cell death triggered by cystine depletion and elevated reactive oxygen species (ROS). BAP1, part of the PR-DUB complex, deubiquitinates histone H2Aub at the SLC7A11 promoter, reducing SLC7A11 expression and cystine uptake, which lowers glutathione biosynthesis. This leads to increased lipid peroxidation and ferroptosis initiation [120]. Without BAP1, cells struggle to initiate ferroptosis [119].

The loss of BAP1 disrupts programmed cell death, including apoptosis and ferroptosis, impairing DNA repair and allowing mutated cells to proliferate uncontrollably, resulting in malignant transformation [119,121].

## 6. Tumor Inflammatory Microenvironment in PM—Role in Tumor Progression

The recruitment of pleural macrophages and inflammatory cells in PM creates a peritumoral inflammatory microenvironment that is crucial in tumor growth, progression, and invasiveness.

The tumor microenvironment (TME) is dynamic and contains various cellular components such as mesothelial cells, stromal cells (fibroblasts and carcinoma-associated fibroblasts), mesenchymal stem cells, endothelial cells, and pericytes. It also contains immune cells like tumor-associated macrophages, dendritic cells, B lymphocytes, CD4+ T-helper cells, CD8+ cytotoxic T cells, and regulatory T cells, all supported by an extracellular matrix [41,122,123].

Interactions among cells in the TME and their communication with tumor cells through extra-vesicles and exosomes are vital for cell activation. This activation triggers the release of pro-inflammatory cytokines, chemokines, and growth factors, including IL-1*β*, IL-6, IL-8, IL-10, IL-12, G-CSF, VEGF, HGF, TGF-*β*, FGF, EGF, and TNF-*α*. These processes promote tumor invasion and metastasis by increasing ROS production, altering extracellular matrix protein secretion (such as fibronectin, MMPs, and collagen), and remodeling the extracellular matrix (ECM) [41,124,125,126].

Several factors contribute to the epithelial to mesenchymal transition (EMT) in PM carcinogenesis. Transforming Growth Factor beta (TGF-*β*) is crucial, as it decreases epithelial markers like E-cadherin and occludin while increasing mesenchymal markers such as fibronectin, smooth muscle actin (SMA), and vimentin in mesothelial cells [41,124]. Additionally, fibroblast growth factor 2 (FGF2) and epidermal growth factor (EGF) can promote a fibroblast-like phenotype with invasive traits and reduced cell adhesion in pleural mesothelioma [41].

These mediators of inflammation and oxidative stress can activate or suppress cellular signaling pathways, promoting tumor advancement, metastasis, and immune evasion, including through the PD-1/PD-L1 pathway [41].

Lee et al. [122] identified two inflammatory tumor microenvironments in PM using mass cytometry: TiME 1 and TiME 2. TiME 1 features dysfunctional CD8+ T cells with partial exhaustion (PD-1+CTLA-4+CD8+ T cells) that respond to checkpoint blockade therapies, enhancing progression-free survival (PFS) [127]. It also includes HLA-DR+ tumor cells producing cytokines like IL-10, IL-6, and TNF-*α*, along with phosphorylated transcription factors such as HIF-1A, CPARP, and STAT3, and plasmacytoid dendritic cells (pDCs) known for their antiviral functions through type I interferon secretion [122].

T regulatory cells (ICOS+-CTLA4+ T-regs) and CXCR4+CD38-CD8+ cells, as well as neutrophils, DCs, and tumor-associated macrophages expressing PD-L1+ (PD-1+TAMs), CAFs, and other cells have been identified in the TiME 2 subtype [122,126]. Some authors consider TiME 1 to be a good TiME, while TiME 2 is a bad TiME [122,126]. Good TiME is associated with a favorable response to immune checkpoint blockade, using inhibitors of immune checkpoint proteins CTLA-4 and PD-1 [122,126].

The PD-L1/PD-1 pathway (PD-L1 (programmed death ligand 1)/PD1 (programmed cell death 1) is important in regulating immune responses, but it also promotes carcinogenesis. T lymphocytes express the checkpoint protein PD-1+, which regulates immune responses and inhibits T cell inflammatory activity, preventing autoimmune reactions. The interaction between PD-1+ lymphocytes and PD-L1+ tumor cells reduce the T lymphocyte’s ability to attack tumor cells. As a result of the overexpression of PD-L1, the tumor can escape the body’s defense system [9].

PD-L1 is more frequently expressed by tumor cells in non-epithelioid compared to epithelioid PM. PD-L1 was associated with a promotion of the T cell infiltration and their activation, but also an increase in Tregs and the expression of the T cell inhibitory markers (e.g., TIM-3 T cell immunoglobulin domain and mucin domain-3) [41].

Brcic et al. [9] found that low PD-L1 expression in tumor cells correlates with better survival rates, while high PD-L1 expression (over 10%) is linked to significantly reduced overall survival, regardless of histology, gender, age, or treatment stage.

This concept provides a promising foundation for developing pharmacological treatments to inhibit the PD-L1/PD-1 pathway. While some agents have shown positive responses in treating skin melanomas and lung cancer [128], further investigation is needed for their use in PM [129].

## 7. Prognosis of Pleural Mesothelioma

Chen et al. [130] analyzed data from 1978 PM patients, reporting an OS rate of 10 months and a cancer-specific survival (CSS) rate of 11 months. They established cut-off values to classify patients into high- and low-risk categories based on factors such as age, gender, histology, insurance status, T stage, M stage, surgical intervention, and chemotherapy, and developed an online dynamic nomogram for predicting patient survival outcomes [131].

Key adverse prognostic factors include sarcomatoid and biphasic variants of PM, male gender, young age, anemia, poor performance status, and elevated lactate dehydrogenase (LDH) levels.

Zhang et al. [44] analyzed the LENT and BRIMS scoring systems as prognostic indicators for PM and malignant pleural effusion (MPE). They found a significant correlation between higher scores and poorer outcomes, including lower survival rates, as well as a link to available treatment options like chemotherapy and surgery.

Preventing exposure is currently the most effective strategy against PM, as shown by Hemminki et al. [8]. Nordic countries, including Denmark, Finland, Norway, and Sweden, pioneered the NORDCAN cancer registry and banned asbestos in the 1980s, enabling the monitoring of PM-related condition declines. However, eliminating asbestos alone is not enough if air pollution continues. Research from South Korea [132] reveals that mortality risk from PM nearly doubles with increased exposure to pollutants like sulfur dioxide (SO_2_) and nitrogen dioxide (N_2_). Despite banning asbestos in 2009, South Korea still sees rising PM incidence due to the long latency period of related diseases.

Despite the poor prognosis of PM, Sayan et al. [133] analyzed the effects of multimodal therapy (MMT)—including chemotherapy, radiotherapy, and surgery—on OS rates. Their findings showed median survival times and five-year survival rates two to three times higher than the control group not receiving MMT. While survival is still measured in months, this offers a glimmer of hope.

## 8. Pleural Mesothelioma Treatments in Development

One of the earliest therapeutic regimens for the oncological treatment of PM involved a platinum-pemetrexed combination, typically administered over an average of six cycles [134]. The outcomes were notably improved with the addition of bevacizumab, a VEGF inhibitor [135]. The adverse effects associated with platinum–pemetrexed were mitigated through the administration of folic acid and vitamin B12 to the patients [134]. Furthermore, platinum-based therapies have been shown to enhance survival rates in patients exhibiting loss-of-function mutations in BAP1 and DNA repair genes, in contrast to those without such mutations [136]. Therefore, it is imperative to achieve an accurate histopathological and molecular diagnosis of the disease to facilitate targeted treatment. In the following sections, we will delve deeper into the targeted therapies for MTAP and BAP1.

Engineered molecular therapies targeting specific biological pathways have proven effective for various neoplasms, but not for PM. Consequently, new therapies are being evaluated in clinical trials. The difficulty in developing effective treatments for PM is due to tumor heterogeneity and the surrounding inflammatory microenvironment.

### 8.1. BAP 1 Target Therapies

BAP mutations are linked to PM carcinogenesis, leading to the investigation of therapies targeting this biomarker. Loss of BAP1 expression in PM correlates with altered histone deacetylase levels, notably increased HDAC1 and decreased HDAC2. Some researchers suggest BAP expression status could predict which patients might benefit from HDAC inhibitors like vorinostat [101]. The VANTAGE 014 phase 3 trial, involving 661 patients, evaluated vorinostat’s effectiveness but found no significant improvement in overall survival compared to placebo [137]. Further research is needed to explore this treatment’s potential for PM [101,119,137].

EZH2 inhibitors offer an alternative treatment strategy, as EZH2 expression is elevated in tumors lacking BAP1 and is linked to poor prognosis [138]. In a phase 2 trial by Zauderer et al. [139], Tazemetostat (Tazverik) was given to 61 patients, but only 2 showed a partial response.

The chromatin-associated PARP enzyme is essential for DNA repair, making cancer cells with defective repair mechanisms targets for PARP inhibitors [101]. Patients with BRCA1 and BRCA2 mutations in breast, ovarian, or pancreatic cancers respond well to this treatment [101]. There is also interest in PARP inhibitors like Olaparib [140] and Niraparib [115,141] for managing PM with BAP1 deletion [88]. Some studies suggest BAP-negative cells are more sensitive to PARP inhibitors [115,142], while others indicate that BAP1 absence does not significantly impact inhibitor response [109,143].

Hassan et al. [136] found that patients with PM and BAP1 mutations are more sensitive to platinum-based chemotherapy, like cisplatin, which causes DNA damage and cell death. Reduced BAP1 expression may also improve tumor response to immunotherapy [144]. BAP-negative tumors have a unique inflammatory microenvironment with increased immune cell infiltration and activated immune checkpoints, suggesting they may respond well to immune-checkpoint inhibitors (e.g., anti-PD-L1, anti-PD-1, and anti-CTLA-4 agents), which show 20–30% response rates in clinical trials [128,145]. Recent studies indicate that combining these inhibitors or using them with chemotherapy may enhance their effectiveness [37].

### 8.2. MTAP-Deleted Tumours—New Target Therapies

The PRMT5 enzyme is involved in protein metabolism, prompting research into PRMT5 inhibitors as potential treatments for malignancies linked to MTAP deletion. MTA accumulation inhibits SAM binding to PRMT5, forming the PRMT5-MTA complex and reducing enzyme activity in MTAP-deficient tumors. New PRMT5 inhibitors aim to disrupt SAM binding or destabilize the PRMT5-MTA complex [73,74,75]. PRMT5 plays a catalytic function, and when SAM is lowered further in MTAP-negative tumors, the tumor cells become even more vulnerable to PRMT5 depletion [75]. Recent research by Smith et al. [146] identified MRTX1719 as a promising candidate that selectively inhibits PRMT5 in MTAP-negative cells by stabilizing the PRMT5-MTA complex. MRTX1719 is taken orally and shows a dose-dependent response, but further studies are needed to assess its efficacy and safety.

Another potential therapeutic target in MTAP-negative tumors would be MAT2A, the main source of SAM, and substrate for PRMT5 [73,74,75]. Kalev et al. [78] demonstrated that MAT2A inhibitors (AGI-24512 and AG-270) significantly lower SAM levels and PRMT5 activity in MTAP- cells, inducing an anti-proliferative action.

Given the metabolic activity of MTAP, a potential therapeutic strategy for MTAP negative tumors would be a combination between MTA and purinic analogues 6′-tioguanine (6TG) and 2′-fluoroadenine (2FA). Tang et al. [79] studied the action of these compounds in vitro as well as in vivo, concluding that the best combination is between 2FA and MTA, with beneficial effect against four different types of human MTAP negative tumor cell lines.

At present, the research activity is primarily focused on assessing the efficacy of various therapeutic agents as second or third-line treatments for PM patients, particularly regarding their impact on survival rates and adverse effects. Additionally, several experimental studies are being investigated but have not yet received regulatory approval Table 1. Due to the significant variability in study conditions and methodologies, a meta-analysis cannot be conducted on these studies. Nevertheless, these studies provide a valuable foundation for future investigations into pharmacological treatments in precision medicine, particularly given that the side effects observed were predominantly mild.

A novel concept in the research community suggests that using multiple FDA-approved drugs with known side effects and pharmacokinetic profiles could improve testing efficiency and shorten clinical trial durations Table 2.

The prevailing approach involves incorporating immunotherapy either as a standalone treatment or as an adjunct to chemotherapy. The most extensively researched antibodies include anti-PD-1, alongside others such as anti-VEGF/VEGFR, anti-CTLA-4, and mesothelin-targeted immunotherapy [147,148,149]. As detailed in Table 1, Durvalumab was evaluated by Forde et al. [150] as a first-line therapy for patients with PM, resulting in a median overall survival (OS) increase to 20.4 months. Additionally, Canova et al. [151] assessed it as a second-line option following tumor recurrence, achieving an OS of approximately 7.3 months. Nivolumab, another PD-1 antibody, has been investigated across multiple studies [148,152,153], both as monotherapy and in conjunction with Ipilimumab [148]. Although Kindler et al. [147] found no statistically significant improvement in patient survival with Anetumab Ravtansine or Vinorelbine, further research is warranted to evaluate therapeutic effectiveness. A case report also noted that the combination of Tislelizumab, an anti-PD1 antibody, and Anlotinib achieved an overall survival of over 10 months [154]. Most patients undergoing immunotherapy experienced only mild and manageable side effects, which is essential for the continued management of these cases; however, there were instances where patients did not respond to corticosteroids. See Table 1.

None of the therapies show exceptional efficacy, but results vary by conditions. For instance, lurbinectedine’s effectiveness is unaffected by BAP1 status [155], while metformin’s impact depends on timing and dosage [156]. Additionally, BAP1 inactivation contributes to gemcitabine resistance, among other factors [149] (Table 2).

Science is in a state of constant evolution, with advancements occurring daily in the development of novel treatments and approaches to patient care for those with PM.


diagnostics-15-01323-t001_Table 1Table 1Treatments tested on patients with PM, as first-line or palliative care.ArticleTreatmentMechanismDoseMedian of Cycles Per PatientNumber of Patients/SamplesLine of TreatmentMedian OSMedian PFSORRSide EffectsObservationAhmadzada 2020 [157]PembrolizumabPD-1 antibody200 mg or 2 mg/kg every 3 weeks6984 vs. 63 vs. 319.5 months4.8 months18%Pneumonitis, nefritis, hepatitis, etc.
Hassan 2019 [136]Platinum-basedDisrupting DNA replication (chemotherapy)

2861st



OS was higher for the patients with BAP1 mutationsLam 2020 [158]AZD4547FGF inhibitor80 mg × 2/day over 3 weeks4241st/2nd7.3 months3 months
Hyperphosphatemia, xerostomia, mucositis, retinopathy, etc.There is no improvement in patient status as a second-line therapy, following treatment with platinum-based chemotherapyZauderer 2021 [159]LY3023414Dual PI3K/mTOR inhibitor200 mg × 2/day
422nd/3rd
2.83 months
Fatigue, nausea, decreased appetite, vomiting, diarrhea, etc.The study took into trial patients with advanced mesothelioma (pleural and peritoneal)Passiglia 2024 [160]Niraparib and DostarlimabPARP-inhibitor and PD-1 antibody

17
4.2 months3.1 months6%Lymphopenia, anemia, hyponatremia, hypokalemia, etc.The study took into trial patients with pleural mesothelioma or NSCLCHearon 2020 [161]PembrolizumabPD-1 antibody200 mg every 3 weeks31



Fatigue, hypothyroidism, lymphopenia, diabetes type I, etc.Case study where the effect of pembrolizumab was durable after the drug was stoppedGhafoor 2021 [162]OlaparibPARP-inhibitor300 mg × 2/day for 3 weeks4232nd/3rd8.7 months3.6 months4%Nausea, renal toxicity, fatigue, etc.The study involved patients with mesothelioma (pleural and peritoneal)Forde 2021 [150]Durvalumab plus platinum–pemetrexedPD-1 antibody and chemotherapy1.120 mg Durvalumab i.v. every 3 weeks
551st20.4 months6.7 months56.40%Fatigue, nausea, anemia, etc.PFS and OS were statistically better than the PFS and OS of platinum-based monotherapyAdusumilli 2021 [163]CAR T cell therapy and pembrolizumabCAR T cell infusion and PD-1 antibody0.3–60 M CAR T cells/kg intrapleural
232nd/3rd23.9 months



Watanabe 2021 [164]AmrubicineInhibition of DNA topoisomerase II35 mg/m^2^ 2 days/week for 3 weeks352nd/3rd9.1 months2.4 months0%Neutropenia, anemia, decreased appetite, constipation, etc.There were no responders to Amrubicine, but an SD (stable disease) was observed in three out of five patientsXie 2022 [165]CrizotinibProtein kinase inhibitor

12nd7.6 YEARS6 YEARS

The patient has MPM positive for CD74-ROS1 fusionKindler 2023 [147]Anetumab Ravtansine vs. VinorelbineAntibody anti mesothelin and inhibitor of mitosisAR: 6.5 mg/kg once over 3 weeks V:30 mg/m^2^ once every week
2482nd9.5 months vs. 11.6 months4.3 months vs. 4.5 months
Neutropenia, pneumonia, dyspnoea, etc.There was no statistically significant difference between the treatmentsFennell 2021 [152]Nivolumab vs. PlaceboPD-1 antibody240 mg every 2 weeks
3322nd10.2 months vs. 6.9 months3 months vs. 1.8 months11% vs. 1%Dyspnoea, pneumonia, lower inspiratory tract infection, etc.95% of the patients had pleural mesothelioma, the rest had peritonealMark 2022 [155]LurbinectineBlocking the cell cycle in the S-phase and activation of the DNA damage response

422nd/3rd11.5 months4.1 months
Viral pneumonitis, dyspnoea, haert failure, etc.The study classified the group into categories by survival and tried to find a connection between OS and their genesZhang 2022 [154]Tislelizumab and AnlotinibPD-1 antibody and tyrosine kinase inhibitor200 mg Tislelizumab/day and 10 mg Anlotinib daily for 2 weeks and one week off.
12nd
10 months (until the article was published)


Canova 2022 [151]DurvalumabPD-1 antibody1500 mg Durvalumab every 4 weeks3692nd7.3 months1.9 months10%Atrial fibrillation, hyper/hypothyroidism, ischemic colitis, diarrhea, etc.
CheckMate 743 [148]Nivolumab and IpilimumabPD-1 antibody and antibody anti CTLA-4 vs. chemotherapyN: 3 mg/kg i.v. once every two weeks and I:1 mg/kg i.v. once every six weeks12 and 4 vs. 6300 vs. 3031st18.1 months vs. 14.1 months6.8 months vs. 7.2 months
Diarrhoea, pruritus, fatigue, hypothyroidism, nausea, etc.OS did not differ between histological types of M while using N and I, but differed dramatically while using chemotherapy; 8.8 months for non-epithelioid vs. 16.5 months for epithelioid.Pinto 2021 [149]Gemcitabine +/− RamucirumabChemotherapy and antibody anti VEGF/VEGFRR: 10 mg/kg once every 3 weeks G: 1000 mg/m^2^7.5 vs. 3.51612nd13.8 months vs. 7.5 months6.4 months vs. 3.3 months
Neutropenia, hypertension, thrombembolism, etc.OS was longer in the gemcitabine plus ramucirumab group than into gemcitabine plus placebo groupYap 2021 [166]PembrolizumabPD-1 antibody200 mg i.v. once every 3 weeks61182nd10 months2.1 months
Colitis, hyponatraemia, pneumonitis, etc.Pembrolizumab has a good antitumor activity, regardless of PD-1 statusCosta 2022 [153]NivolumabPD-1antibody3 mg/kg once every 2 weeks
12nd


Arthralgia
Szlosarek 2023 [167]Pegargiminase and ChematherapyArginine deprivation therapy36.8 mg/m^2^ i.m. once per week
2491st9.3 months6.2 months

The study was conducted with non-epithelioid pleural mesothelioma patients.
diagnostics-15-01323-t002_Table 2Table 2Experimental therapies in PM.ArticleTreatmentMechanismDoseNumber of SamplesObservationAnobile 2021 [168]LurbinectedinBlocking the cell cycle in S-phase and activation of the DNA damage response0.07–4.5 nM12Efficacy independent of the BAP1 statusBorchert 2019 [142]OlaparibPARP-inhibitor1–10 μm90DDB2 and RAD50 are associated with long survival if given OlaparibGuazzelli 2019 [169]GemcitabineDisrupting DNA replication (chemotherapy)0.1–50 μm
Inactivation of BAP1 determines resistance to gemcitabineKumar 2019 [170]Vinorelbine or Mitomycin, vinblastine, or cisplatinInhibition of mitosis because of the interaction of tubulin (chemotherapy)
60OS was no different between the treatment armsSalaroglio 2022 [171]MLN4924 +/− cisplatin or placeboSelective NEDD8 inhibitor and chemotherapy5 mg/kg cisplatin i.p. once a week; 25 mg/kg MLN4924 s.c. 5 days/week40 miceThese two drugs have a synergic anti-tumor effect, independent from the MPM histotypeRossini 2021 [156]MetforminStimulates the apoptotic process, associated with decreased Notch1 activation1–50 mM
Metformin succeeded to inhibit cell viability of PM; dose and time dependent


## 9. Conclusions

Pleural mesothelioma continues to pose significant challenges within the medical community due to its complex diagnostic process and unfavorable prognosis. While prevention plays a crucial role in managing this condition, numerous countries still lack adequate regulations concerning asbestos exposure and air quality. Given the global mobility of individuals for work and travel, it is imperative for healthcare professionals to be equipped to address this disease, which is anticipated to reach its peak incidence in the current decade.

BAP1 and MTAP genes are significantly involved in the development of cancer. The application of immunohistochemistry techniques to assess these markers is likely to enhance the quality-to-cost ratio of diagnostic procedures. BAP1 and MTAP engage in numerous complex pathways that interact with various cells and the surrounding microenvironment, each presenting potential avenues for therapeutic intervention. While chemotherapy continues to be the standard treatment approach, there is an increasing exploration of alternative substances as second- or third-line therapies. Notably, some of these agents, including Metformin, are widely used, thereby facilitating the advancement of clinical trials and their subsequent integration into standard medical practice.

It is essential to gain a deeper understanding of the genes and markers involved in the pathological processes to provide optimal solutions for our patients.

## Figures and Tables

**Figure 1 diagnostics-15-01323-f001:**
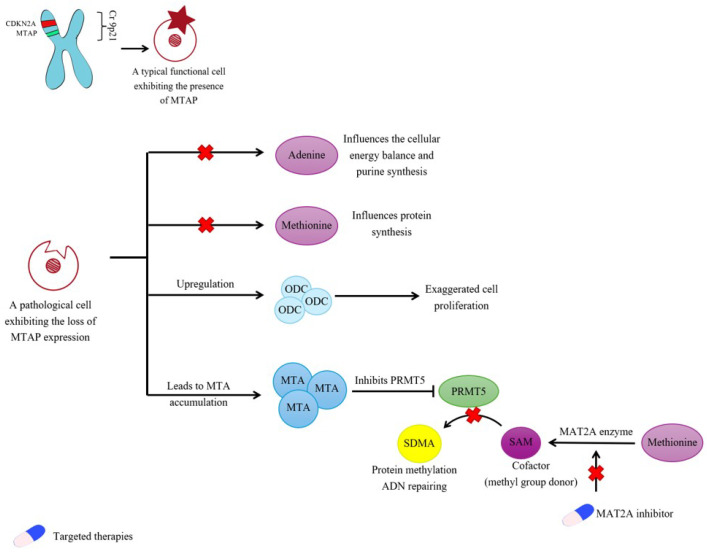
The implications of the loss of MTAP expression and the potential for targeted therapeutic drugs. MTAP—methylthioadenosine phosphorylase; ODC—Ornithine Decarboxylase; PRMT5—protein arginine methyltransferase 5; MAT2A—metabolic enzyme methionine adenosyltransferase II alpha; SAM-S-adenosyl-l-methionine; SDMA—symmetric demethylation of arginine.

**Figure 2 diagnostics-15-01323-f002:**
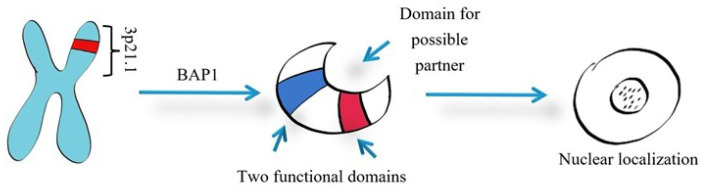
BAP1 schematic structure; BRCA1-associated protein 1 (BAP1).

**Figure 3 diagnostics-15-01323-f003:**
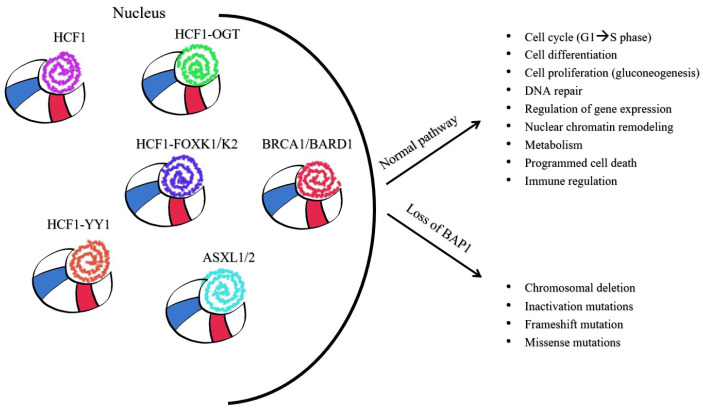
The role of BAP1 protein complexes in cell regulation and death: HCF1—host cell factor 1; BRCA1—breast cancer gene 1; BRCA1-associated protein 1 (BAP1); BARD1—BRCA1 associated RING domain protein 1; ASXL1/2—additional sex comb like 1 or 2; OGT—O-linked N-acetylglucosamine transferase; YY1—Ying Yang 1 transcriptional repressor; FOXK1/2 forkhead transcription factors.

**Figure 4 diagnostics-15-01323-f004:**
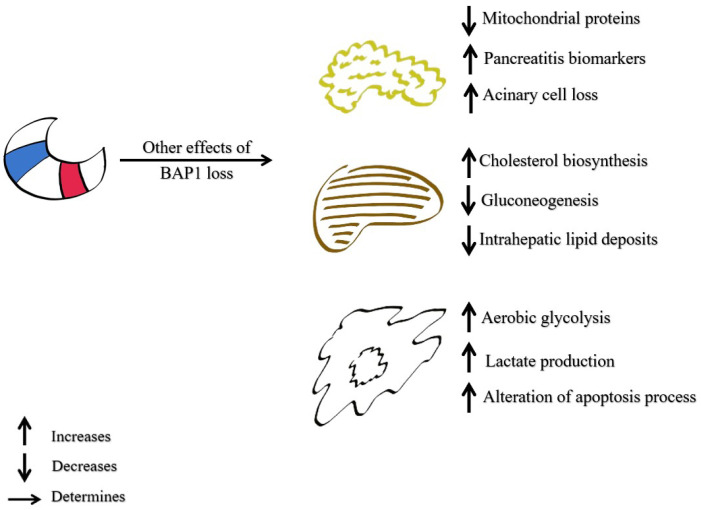
Role of BAP1 loss in metabolism; BRCA1-associated protein 1 (BAP1).

## Data Availability

All data can be found in the references.

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
