# Peer review of "Molecular Insights into Pleural Mesothelioma: Unveiling Pathogenic Mechanisms and Therapeutic Opportunities"

_diagnostics, 2025, doi:10.3390/diagnostics15111323_

Round 1
Reviewer 1 Report
Comments and Suggestions for Authors
Dear Authors,
I have read a review about pleural mesothelioma (PM), especially on the molecular insights. There are several points that I would like to highlight:
1) The scope is too broad. The authors attempted to delineate general aspects as well as molecular insights, which resulted in the manuscript being less focused on either aspect. As the general aspects of PM can be too broad, I suggest removing this part from the title and the aim, while keeping the other subheaders as it is
2) Lines 30-38 are not suitable in the introduction section. This section should highlight the urgent need for an updated review of PM
3) What is a para-professional exposure?
4) Regarding point number 1, the authors aim to highlight the general aspects of PM but did not elaborate on what the symptoms of PM are, brushing them off as "non-specific". The authors should delineate what the symptoms are, and can explain that these symptoms are non-specific
5) What do the authors mean by "primary diagnostic tool"? Is it the first-line imaging modality or is it the reference standard? The authors should explain what can be found from the laboratory work, and what imaging findings can be expected from chest x-ray, ultrasound, CT, and MRI, since the authors want to discuss the "general aspects" of PM
6) There should be better annotations in each Figure. For example, in Figure 1, there are two types of arrows; what does each arrow mean? Does increased MTA accumulation and excretion inhibit SDMA or the PRMT5?
7) Line 237, the authors cited reference number [63]. Although the reference mentioned the kappa coefficient to be 0.8, they also mentioned that "Although, the results of MTAP
IHC do not show perfect conformity with 9p21 FISH, the underlying reason for these
differences is biological variation rather than technical flaws in procedures. It is
important to note that the MTAP loss of expression by IHC is not directly indicative of
9p21 HD; it is rather a surrogate for co-deletion of MTAP with 9p21 HD". Please include these sentences in the discussion as well
8) Subheader 5.2, "BAP1 roles is pathogenesis" does not make sense. Do the authors mean "in"?
9) Line 314, the authors cited reference number [62], where the authors put this citation for the sentence "..diagnostic sensitivity....75% for the epitheliod variant.."
Citation number 62 is a review article, which in turn cites this article
"Farzin M, Toon CW, Clarkson A, Sioson L, Watson N, Andrici J, et al. Loss of expression of BAP1 predicts longer survival in mesothelioma. Pathology. 2015;47(4):302-7."
In the original article, the authors did not mention anything about sensitivity. This is the biggest issue in this review article, where the authors cite a review without cross-checking the primary source. I only included two examples (this point and number 7), where there are plenty of other examples as well. I hope the authors read each article carefully before resulting in a prolonged peer-review process
10) Subheader 5.2.2. and 5.2.3., why do the BAP1 has a "-" annotation? Should it be BAP1 only or BAP1-?
11) Please provide a proper citation for reference 126
12) The authors jump straight into developing therapies without discussing the current available treatment regimen, which should be discussed, as the authors would like to discuss "general aspects" of PM
13) The authors discuss MTAP in the potential therapy section without discussing it in the previous sections. Please address this point accordingly.
Author Response
Comment 1: The scope is too broad. The authors attempted to delineate general aspects as well as molecular insights, which resulted in the manuscript being less focused on either aspect. As the general aspects of PM can be too broad, I suggest removing this part from the title and the aim, while keeping the other subheaders as it is
Response 1: We appreciate your feedback and have revised the title as follows „Molecular insights into in pleural mesothelioma: unveiling pathogenic mechanisms and therapeutic opportunities”
Comment 2: Lines 30-38 are not suitable in the introduction section. This section should highlight the urgent need for an updated review of PM.
Response 2: We have eliminated lines 30 through 38 and incorporated a new paragraph following line 140-154. The newly added paragraph is provided below and has been highlighted in the manuscript.
“Histopathological analysis identifies three types of PM with prognostic significance: epithelioid (60%), sarcomatoid (20%), and biphasic (20%). Each type has subtypes based on architectural patterns (tubulopapillary, trabecular, adenomatoid, solid, micropapillary) and cytological features (rhabdoid, deciduoid, small cell, clear cell, signet ring, lymphohistiocytoid, transitional, pleomorphic), with stromal characteristics including myxoid, desmoplastic, and heterologous differentiation. The latest WHO Classification of Pleural Tumors (Sauter 2022) defines biphasic mesothelioma as having both epithelioid and sarcomatoid patterns, with each component needing to be at least 10% in resection specimens, regardless of their proportions in small samples. Furthermore, the line between diffuse and localized mesotheliomas is well established, because the latter has a better prognosis. Alongside the latest classification, a new entity, mesothelioma in situ, has been added in the Benign and preinvasive mesothelial tumors subsection, identified by loss of BAP1 (BRCA-Associated protein 1) and/or MTAP (methylthioadenosine phosphorylase) expression by immunohistochemistry (IHC) and/or p16/CDKN2A homozygous deletion detected by FISH (fluorescence in situ hybridization) (Sauter 2022, Dacic 2022)”
Comment 3: What is a para-professional exposure?
Response 3: Individuals in close proximity to employees who have been exposed to asbestos may experience indirect effects from this mineral through the inhalation of particles present on the workers' belongings or clothing. This situation exemplifies para-professional exposure. An additional clarification has been incorporated into the manuscript, located at line 42-44.
Comment 4: Regarding point number 1, the authors aim to highlight the general aspects of PM but did not elaborate on what the symptoms of PM are, brushing them off as "non-specific". The authors should delineate what the symptoms are, and can explain that these symptoms are non-specific
Response 4: We have incorporated the information. The subsequent paragraph is located at line 115-121 „Exposure to asbestos can result in various clinical symptoms, including unilateral pleurisy, dry cough, hemoptysis, dysphagia, night sweats, clubbing of fingers, ascites, superior vena cava syndrome, Horner's syndrome, laryngeal nerve paralysis, and paraneoplastic syndrome, among others. Many of these symptoms develop progressively and may mimic malignant infiltration. Given their non-specific characteristics, these symptoms do not serve as diagnostic criteria for pleural mesothelioma, even in cases of confirmed asbestos exposure. (Scherpereel2020 , Mizuhashi2021, Brims2021, Cotran1998, Yalcin2013, Murali2010)”
Comment 5: What do the authors mean by "primary diagnostic tool"? Is it the first-line imaging modality or is it the reference standard? The authors should explain what can be found from the laboratory work, and what imaging findings can be expected from chest x-ray, ultrasound, CT, and MRI, since the authors want to discuss the "general aspects" of PM
Response 5: Thoracic CT, whether native or contrast-enhanced, is primarily employed when patients exhibit respiratory symptoms and notable decline in health, as the procedure requires only a brief duration—sufficient for breath-holding. Typically, this results in high-resolution images that facilitate diagnosis, subsequently validated through biopsy. Consequently, CT is regarded as the primary diagnostic tool.
Comment 6: There should be better annotations in each Figure. For example, in Figure 1, there are two types of arrows; what does each arrow mean? Does increased MTA accumulation and excretion inhibit SDMA or the PRMT5?
Response 6: We have revised Figures 1 and 4 accordingly.
Comment 7: Line 237, the authors cited reference number [63]. Although the reference mentioned the kappa coefficient to be 0.8, they also mentioned that "Although, the results of MTAP IHC do not show perfect conformity with 9p21 FISH, the underlying reason for these differences is biological variation rather than technical flaws in procedures. It is important to note that the MTAP loss of expression by IHC is not directly indicative of 9p21 HD; it is rather a surrogate for co-deletion of MTAP with 9p21 HD". Please include these sentences in the discussion as well.
Response 7: The paragraph has been amended, and the subsequent section is located at line 238-241 „It is crucial to recognize that the primary cause of these discrepancies is not due to technical deficiencies, but rather biological variations. The study effectively concluded that the loss of MTAP serves as a more appropriate surrogate than a direct substitute.(Hamasaki2019)”
Comment 8: Subheader 5.2, "BAP1 roles is pathogenesis" does not make sense. Do the authors mean "in"?
Response 8: Yes, unfortunately it was a typo, we have rectified it. Thank you.
Comment 9: Line 314, the authors cited reference number [62], where the authors put this citation for the sentence "..diagnostic sensitivity....75% for the epitheliod variant.."
Citation number 62 is a review article, which in turn cites this article
"Farzin M, Toon CW, Clarkson A, Sioson L, Watson N, Andrici J, et al. Loss of expression of BAP1 predicts longer survival in mesothelioma. Pathology. 2015;47(4):302-7."
In the original article, authors did not mention anything about sensitivity. This is the biggest issue in this review article, where the authors cite a review without cross-checking the primary source. I only included two examples (this point and number 7), where there are plenty of other examples as well. I hope the authors read each article carefully before resulting in a prolonged peer-review process.
Response 9: We have reanalized the article in question, and observed that the citing refers only to this paragraph „Loss of BAP1 expression has been associated with younger age at onset and improved median survival in MPM, although BAP1 expres-sion as a prognostic biomarker remains controversial .” We could not find the citation for the diagnostic sensitivity data, consequently, we substitute the data and citation as outlined. This can be located at line 323-329.
„The diagnostic sensitivity of BAP1 for pleural mesothelioma exhibits significant variation across histological subtypes, ranging from 56% to 81% for epithelioid mesothelioma and from 0% to 63% for sarcomatoid mesothelioma(Kinoshita2020.2). Furthermore, the specificity of BAP1 approaches 100% in both cytological and biopsy specimens across all three histological subtypes. It is important to note that acquiring sarcomatoid-type cells from pleural fluid is particularly challenging, and studies typically involve cohorts with a limited number of sarcomatoid subtype cases. (Berg 2019, Chapel2022, Hida 2017, Kinoshita 2017,Kinoshita2018, Lynggard 2022, Berg 2019, Terra 2021, Hiroshima2020).”
Comment 10: Subheader 5.2.2. and 5.2.3., why do the BAP1 has a "-" annotation? Should it be BAP1 only or BAP1-?
Response 10: It should solely be BAP1. We utilized the '-' annotation in the same manner as one might use the ':' annotation. The annotation has been eliminated and reworded.
Comment 11: Please provide a proper citation for reference 126
Response 11: Updated citation to : Dynamic Nomogram. (n.d.). Shinyapps.Io. Retrieved May 11, 2025, from https://mpmsurvival.shinyapps.io/MPMforOS/
Comment 12: The authors jump straight into developing therapies without discussing the current available treatment regimen, which should be discussed, as the authors would like to discuss "general aspects" of PM
Response 12: We have attempted to encapsulate the existing treatment regimen in a concise manner, which can be found at line 541-551 „One of the earliest therapeutic regimens for oncological treatment of PM involved a platinum-pemetrexed combination, typically administered over an average of six cycles (Volgelzang 2003). The outcomes were notably improved with the addition of bevacizumab (Zalcman 2016). The adverse effects associated with platinum-pemetrexed were mitigated through the administration of folic acid and vitamin B12 to the patients (Volgelzang 2003). Furthermore, platinum-based therapies have been shown to enhance survival rates in patients exhibiting loss-of-function mutations in BAP1 and DNA repair genes, in contrast to those without such mutations (Hassan 2019). Therefore, it is imperative to achieve an accurate diagnosis of the pathology to facilitate targeted treatment. In the following sections, we will delve deeper into the targeted therapies for MTAP and BAP1.”
Comment 13: The authors discuss MTAP in the potential therapy section without discussing it in the previous sections. Please address this point accordingly.
Response 13: We appreciate your feedback and have incorporated the following paragraph at the line 280-285 .”Consequently, the primary area of scientific inquiry at present is the PRMT5-MTA complex and the potential methods for its inhibition (Kalev 2021). Given its involvement in metabolic processes, there are numerous potential targets for pharmacological intervention, prompting the investigation of pharmaceutical combinations that incorporate both MTA inhibitors and purine analogs (Tang 2018). Further elaboration will be provided in the section dedicated to targeted therapies.”
Reviewer 2 Report
Comments and Suggestions for Authors
Dear Authors, thank you for this opportunity to review this manuscript. First of all, I would like to congratulate the authors for a well-designed review. The authors seek new predictive and prognostic biomarkers in pleural mesothelioma. The authors examined the immunological network signature of naïve non-oncogene-directed non-small cell lung cancer patients treated with anti-PD1 therapy. The authors focus on the mutations and possible pathomechanisms underlying the development of pleural mesothelioma. At the end of the article, current therapeutic options are summarized. The analysis seems well-designed, with a clear discussion and short conclusions. The literature is up to date.
My only comment on the article is that immunotherapy is now available for patients with mesothelioma, both in the first-line setting and in the second or third-line setting, and I would therefore recommend that it be discussed in more detail in the final part of the article.
Author Response
Comment 1: My only comment on the article is that immunotherapy is now available for patients with mesothelioma, both in the first-line setting and in the second or third-line setting, and I would therefore recommend that it be discussed in more detail in the final part of the article.
Response 1: We sincerely appreciate your feedback. In the concluding section, we have endeavored to elaborate further on immunotherapy. You can locate this information at line 617-633 „The prevailing approach involves incorporating immunotherapy either as a standalone treatment or as an adjunct to chemotherapy. The most extensively researched antibodies include anti-PD-1, alongside others such as anti-VEGF/BEGFR, anti-CTLA-4, and anti-mesothelin (Pinto 2021, CheckMate 743, Kindler 2023). As detailed in Table 1, Durvalumab was evaluated by Forde et al. (Forde 2021) as a first-line therapy for patients with PM, resulting in a median overall survival (OS) increase to 20.4 months. Additionally, Canova et al. (Canova 2022) assessed it as a second-line option following tumor recurrence, achieving an OS of approximately 7.3 months. Nivolumab, another PD-1 antibody, has been investigated across multiple studies (Costa 2022, CheckMate 743, Fennel 2021), both as monotherapy and in conjunction with ipilimumab (CheckMate 743). Although Kindler et al. (Kindler 2023) found no statistically significant improvement in patient survival with Anetumab Ravtansine or Vinorelbine, further research is warranted to evaluate therapeutic effectiveness. A case report also noted that the combination of Tislelizumab and Anlotinib achieved an overall survival of over 10 months (Zhang 2022). Most patients undergoing immunotherapy experienced only mild and manageable side effects, which is essential for the continued management of these cases; however, there were instances where patients did not respond to corticosteroids. Table 1.”
Round 2
Reviewer 1 Report
Comments and Suggestions for Authors
The authors have addressed all my issues